# Estimating Destination of Bus Trips Considering Trip Type Characteristics

Soongbong Lee [1], Jongwoo Lee [1], Bumjoon Bae [2], Daisik Nam [3] and Seunghoon Cheon [1,*]

1 Big Data Platform and Data Economy, The Korea Transport Institute, 370 Sicheong-daero, Sejong 30147, Korea; habanera82@koti.re.kr (S.L.); jongwoo0214@koti.re.kr (J.L.)
2 Center for Privately-Financed Highway Studies, The Korea Transport Institute, 370 Sicheong-daero, Sejong 30147, Korea; bbae2016@koti.re.kr
3 Graduate School of Logistics, Inha University, Incheon 22212, Korea; namd@inha.ac.kr
* Correspondence: sh1000@koti.re.kr

**Abstract:** Recently, local governments have been using transportation card data to monitor the use of public transport and improve the service. However, local governments that are applying a single-fare scheme are experiencing difficulties in using data for accurate identification of real travel patterns or policy decision support due to missing information on alighting stops of users. This policy limits its functionality of utilizing data such as accurate identification of real travel patterns, policy decision support, etc. In order to overcome these limitations, various methods for estimating alighting stops have been developed. This study classifies trips with missing alighting stop information into trip four types and then applies appropriate alighting stop estimation methodology for each trip type in stages. The proposed method is evaluated by utilizing transportation card data of the Seoul metropolitan area and checking the accuracy for each standard of allowable error for sensitivity analysis. The analysis shows that the stage-by-stage estimation methodology based on the trip type proposed in this study can estimate users' destinations more accurately than the methodologies of previous studies. Furthermore, based on the construction of nearly 100% valid tag data, this study differs from prior studies.

**Keywords:** public transit transaction data; estimation of destination; categorization of trip types; trip chain; travel pattern; historical travel data





## 1. Introduction

Recently, local governments have been using big data in various ways to improve the efficiency and service of transportation operations. In the case of public transportation especially, it is possible to analyze the usage status based on data because most users use transportation cards. In fact, they are used in various areas such as the adjustment of public transport routes and headway of vehicles, usage checking and fare payment, and establishment of a plan for adopting new demand-responsive transport (DRT).

For the effective monitoring of the public transit operation and improvement of the service utilizing transportation card data, it is imperative to identify the travel patterns accurately, including the travel origin and destination of the users. In several cases, however, users do not tag the transportation card at the end of the trip because several local governments apply a single-fare scheme; hence, the information about the alighting stops is missing, and there are difficulties in data utilization, such as identifying the actual travel patterns of the users, monitoring the status, and improving the service based on data.

To address these challenges, various studies have been conducted in South Korea and other countries to estimate the alighting stops when the alighting tag information of users is missing. Although there may be differences in the behavior of omission of alighting stops due to differences in public transportation fare systems by country, the type of data collected and the approach for estimating alighting stops are similar. In particular,

there are several studies focusing on alighting stop estimation methodologies based on the connectivity between trips, i.e., under the assumption that the next trip starts near the destination of this trip [1–18]. Furthermore, studies on alighting stop estimation by applying machine learning methods and analyzing travel patterns based on the past travel history data have been conducted [12–16]. However, relatively few studies have been conducted on the estimation of more complete travel pattern information in public transit systems by combining various alighting stop estimation methodologies.

The main purpose of this study is to propose a method for estimating alighting stops to build reliable OD data so that public transportation card data can be used in actual fields. In particular, it focused on improving the accuracy of estimating alighting stops and inferring alighting stops for all trips. The present study, estimated travel patterns of public transport users more accurately and completely, by comprehensively considering the characteristics of various alighting stop estimation methodologies and the characteristics of the trips for which alighting tag information is missing; this was to contribute to more effective public transport operation monitoring and policy decision-support of local governments. Accordingly, we reviewed various alighting stop estimation methods found in previous studies. We considered the characteristics of each method to define the types of travel with missing alighting stop information and set the stages for the alighting stop estimation. The accuracy of the alighting stop information estimation methodology was verified using the data from the Seoul metropolitan area, which includes alighting tag information for most of the trips, and the accuracy and valid data ratio were analyzed based on the results estimated after completely deleting the alighting stop information. Furthermore, we derived implications by examining the accuracies of alighting stop estimation for each stage and for each standard of allowable error, respectively.

## 2. Literature Review

### 2.1. Methodology for Estimating Destination Based on Trip Chain

A trip-chain-based estimation methodology has been used as the most basic method of estimating alighting stops when alighting tag information is missing. Basically, trip-chain-based alighting stop estimation studies assume that the passenger travels many times consecutively and starts the next trip around the destination of the previous trip. Based on this assumption, they estimate alighting stops when there are two or more boarding records per day for the same card ID [1–5]. Some studies assume that the passenger returns to the resident in the case of the last trip on the analyzed day, while the origin of the first trip on a pertinent day is the resident [7–11]. In some other cases, studies look at the origin of the first trip on the next day to estimate the alighting stop of the final trip on that day [12,13]. Studies have also been conducted for a trip-chain-based alighting stop estimation methodology by considering public transport routes (railways and buses) [14,15].

Some studies validated the estimation results to improve the success rate and accuracy of alighting stop estimations and analyzed major parameters and sensitivity. Specifically, studies were conducted on the validation of trip-chain-based alighting stop estimation results [16,17]; sensitivity analysis for "allowed walking distance" to determine the continuity between two trips as a major parameter when estimating the alighting stop based on the trip-chain [9,10]; determining whether the passenger transferred and analyzing the walking journey time in terms of the trip-chain-based estimation [11,18]. The trip-chain-based alighting stop estimation method is applicable only when the trip-chain can be constructed, and cannot be applied to a single trip.

### 2.2. Methodology for Estimating Destination Based on Pattern Using Historical Data

Studies on various techniques, including artificial intelligence (AI) and probability models, have been conducted to estimate the alighting stops for trips with missing alighting stop information, when the trip-chain-based alighting estimation is impossible because of lack of information on the next trip. Regarding analysis-based studies for the past

travel history, a study analyzed the distribution of the boarding time of the trip for each passenger, and estimated the alighting stops by assuming that the usual boarding stop in a similar time band as the estimation-target trip was the origin, while the usual boarding stop in a different time band was the destination [19]; another study analyzed the past travel history of a similar card ID to estimate the alighting stop based on the similarity to the estimation-target trip [20]. Moreover, a study estimated alighting stops by applying a three-dimensional latent Dirichlet allocation model [21], and another study analyzed major trip origins, destinations, and transfer locations by spatially clustering the locations of the stops and estimated alighting stops [20]. Other studies estimated alighting stops by developing machine learning models and deep learning models [22–25].

### 2.3. Methodology for Estimating Destination Based on Combination of Approaches

If a single estimation methodology, such as a trip-chain-based or AI/probability model-based alighting stop estimation methodology, is applied, there is the limitation that there are trips that are impossible to estimate using a single methodology. It is not possible to estimate alighting stops using trip-chain-based method for trips not linked to other trips. Application of the pattern-based method is not possible without historical trip data. To overcome this limitation, studies have been conducted by applying several alighting stop estimation methods. For example, a study applied the trip-chain-based estimation methodology first and then applied a boarding time similarity-based alighting stop estimation method [19]; a study applied different alighting stop estimation methods by classifying the trip type based on the number of trips per day, the last trip of the day, the transfers in the trip, etc. [26]; a study derived a higher success rate of the alighting stop estimation by applying the trip-chain-based alighting stop estimation methodology in the first stage and the machine learning-based methodology in the second stage [25].

### 2.4. Implication

Table 1 shows the various methodologies and accuracies from the destination inference studies. To date, various methodologies have been developed and applied to estimate alighting stops, including the trip-chain-based estimation and AI/probability model-based estimation methodologies. Research has been advanced to validate the effectiveness and accuracy of the methods, and improve the methodologies to facilitate more accurate alighting stop estimation for more trips. However, when a single methodology is applied, there is the limitation that the scope of the applicable trips is limited [19,25,26]. This study proposes a methodology for estimating the alighting stop in stages, by considering the characteristics and limitations of the trip-chain-based alighting stop estimation and past history data-based alighting stop estimation methodologies. This study aims to build more accurate and complete data for the travel patterns of the public transport users for monitoring and decision-support of public transport operations. Accordingly, we classify trip types by considering the continuity and recurrence of trips based on the trip history data, and applying an appropriate alighting stop estimation method for each type.

In order to improve the accessibility of cities and the efficiency of traffic flow, the connection with infrastructure and services surrounding public transportation is an important factor [27–29]. In particular, in order to implement MaaS (Mobility as a Service) through connection with parking and bike sharing systems, it is essential to estimate accurate alighting stops for public transportation.

**Table 1.** Accuracy of destination inference results from previous studies.

| Researcher | Method | Accuracy | Criteria for Evaluating Accuracy | Reference |
|---|---|---|---|---|
| Li et al. (2011) | Trip-chain | 75–85% | Accuracy of OD Matrix | [6] |
| Ma (2013) | Trip-chain | 91.3–94.6% | Within 2 stops from actual destination | [8] |
| Munizaga et al. (2014) | Trip-chain | 84.2% | - | [16] |
| He et al. (2015) | Historical trip data analysis | 79.2% | Within 400 m from actual destination | [20] |
| Alsger et al. (2016) | Trip-chain | 86.6% | Within 800 m from actual destination | [17] |
| Shin et al. (2016) | Trip-chain | 82.4% | Inferred and actual destinations located in same zone | [5] |
| Kim and Lee (2017) | Trip-chain | 93.6–94.0% | Within 2 stops from actual destination | [10] |
| Kim et al. (2018) | Trip-chain | 90.5% | Within 2 stops from actual destination | [13] |
| Yoo et al. (2019) | Trip-chain and historical trip data analyses | 67.2% | Within 2 stops from actual destination | [26] |
| Lee (2019) | Trip-chain and Gaussian Mixture Model | 86.1% | Within 1 stop from actual destination | [19] |
| Yan et al. (2019) | Trip-chain and machine learning | 74.43% | Inferred and actual destinations located in similar zones | [25] |
| Shin (2020) | Spatial clustering | 80.0% | Within 2 stops from actual destination | [22] |
| Assemi et al. (2020) | Neural network | 86.4% | Within 400 m from actual destination | [24] |

Note: The accuracy of trip destination inference refers to the percentage of trips where the inferred destination matches the actual destination.

## 3. Data

The data utilized in this study contain bus and subway boarding and alighting information, and comprise card ID, boarding stop and time, alighting stop and time, used route, etc. We utilized transportation card data for the whole year of 2020 to estimate the transportation card-based alighting stop considering the characteristics of the trip types. However, when a passenger boarded a bus and utilized a subway on the next trip, we fabricated the data utilizing the location coordinates of the stop to construct the trip-chain. The total number of data (based on buses) utilized in the analysis was 4,110,205,036 (approximately 4.1 billion/year), out of which approximately 17% had no alighting tag information (see Table 2). However, when we examine for each city or province, the percentage of the missing alighting tag data in the remaining regions after excluding the Seoul metropolitan and Daejeon & Sejong areas is approximately 63%, which limits utilizing the data for the analysis of real travel patterns in the public transportation. Therefore, alighting stop estimation is imperative.

**Table 2.** Result of analysis on the status of alighting non-tag by city and province based on traffic card data (April 2020).

| Region | Total Number of Data | Number of Alighting Non-Tag Data | Ratio of Alighting Non-Tag Dat |
|---|---|---|---|
| Seoul | 136,502,534 | 3,933,416 | 2.90% |
| Busan | 37,366,459 | 21,420,213 | 57.30% |
| Incheon | 25,191,574 | 780,977 | 3.10% |
| Gwangju | 8,365,264 | 5,823,725 | 69.60% |
| Daejeon | 10,800,434 | 1,691,237 | 15.70% |
| Ulsan | 6,753,704 | 4,767,535 | 70.60% |
| Sejong | 1,590,931 | 154,489 | 9.70% |
| Gyeonggi-do | 124,531,413 | 1,588,267 | 1.30% |
| Gangwon-do | 3,247,020 | 2,058,875 | 63.40% |
| Chungcheongbuk-do | 4,317,692 | 2,478,044 | 57.40% |
| Chungcheongnam-do | 4,695,978 | 1,658,766 | 35.30% |
| Jeollabuk-do | 4,700,207 | 3,455,147 | 73.50% |
| Jeollanam-do | 3,916,794 | 2,863,507 | 73.10% |
| Gyeongsangbuk-do | 4,823,337 | 3,902,572 | 80.90% |
| Gyeongsangnam-do | 13,049,527 | 9,990,513 | 76.60% |
| Jeju | 4,248,946 | 1,434,272 | 33.80% |
| total | 394,101,814 | 68,001,555 | 17.25% |

## 4. Methodology

### 4.1. Definition of Trip Type

This study defines a trip as the time of boarding a bus to the time of alighting the bus. Furthermore, we check whether the alighting stop information is missing for individual trips, and regarding trips with missing alighting tag information, we classify the trip type (continuous travel, non-continuous recurrent travel, non-continuous non-recurrent travel, route pattern) in terms of the trip continuity and recurrence, based on the identifiable information of other trips of the pertinent card ID. For this, it is presumed that a similar card ID is maintained for the entire time range of the utilized data. Subway trips were excluded from the estimation as passengers are required to tag their transportation cards at the alighting stations, resulting in an alighting tag ratio close to 100%.

To determine the trip type, we built the basic DB utilizing the boarding and alighting information in the sequential order of time for all trips on the pertinent day, and the first trip on the next day for each ID. Regarding the first trip on the next day, the data were constructed for estimation in the event that the passenger did not tag the card when alighting the bus on the pertinent day.

For continuous travel, trip-chain-based alighting stop estimation methodology is applied, based on the assumption that the passenger boards a bus around the position where the *i* route, *j*-th alighting stop information is missing. In other words, a trip must occur on the same day or the next day after the alighting tag information is missing, and an estimation-target route must pass through the boarding stop of the next trip within the allowed walking distance. If all these conditions are met, the trip is considered to have occurred continuously and is classified as a "continuous travel"; otherwise, it is classified as a "non-continuous travel."

The concepts of continuous travel and transfer are different. A trip is defined as continuous travel if a passenger utilized two bus routes by transferring once in the middle, while traveling from the origin to the destination. Furthermore, if the passenger travels

from A to B without transferring, then starts the next trip at B or in the vicinity (within the allowed walking distance) on the same day or the next day, the trip is also defined as continuous travel.

Regarding non-continuous travel, because the alighting stop is estimated by searching the past travel history and finding a trip that is determined to have the similarity temporally and spatially to the trip with missing alighting tag, it is selected by utilizing the card ID's past travel history DB. A trip with temporal similarity refers to a case of boarding a certain number of times (three times per week) or more at the same stop at a similar time, and because the trip occurs repeatedly, it is classified as a "recurrent travel." For example, if a passenger goes to work by bus from A to B (trip 1), utilizes a means other than public transportation to go home from B to A after work (no travel record), and utilizes a bus to go to work from A to B on the next day (trip 2), then trip 1 and trip 2 are non-continuous travels. However, if the trip for commuting from A to B is repeated severally, it is classified as a "recurrent travel."

A trip without even the temporal similarity is classified as "non-continuous," "non-recurrent" travel, and the alighting stop is estimated based on the spatial similarity. In this study, if there is no past travel history of the card ID for the analyzed period, we utilize the boarding and alighting patterns between the stops of the route to estimate the alighting stop.

*4.2. Method*

This study estimates the missing alighting stop by applying an appropriate method for each type of travel with missing alighting stop information, which was classified above. First, regarding "continuous travel," the trip-chain-based alighting stop estimation method is applied, which estimates the missing alighting stop based on the boarding stop information of the next trip. Second, for the "recurrent travel" among "non-continuous travels," we analyze the past travel history information to study the travel pattern for each card ID, and then analyze the potential residence/workplace, based on which the missing alighting stop is estimated. Third, for the "non-recurrent travel," we extract the history of boarding around the boarding stop of the estimation-target trip from the past travel history information and estimate the alighting stop based on the destination information in the past travel history. Finally, if there is no continuous travel or past travel history, the alighting stop is assigned probabilistically based on the route pattern. Table 3 presents the utilized data for each stage. Travel types 1 to 3 use historical data by the individual, and travel type 4 uses historical data by route.

**Table 3.** Utilization data for estimating destination of bus trip by travel type.

| Travel Type | 1. Continuous Travel | 2. Non-Continuous Recurrent Travel | 3. Non-Continuous Non-Recurrent Travel | 4. Route Pattern |
|---|---|---|---|---|
| Input data | The day and tommorow (per day) | Individual potential residence/workplace data (per month) | Individual OD travel pattern data (per year) | Route OD travel pattern data (per year) |
| Data for validation | April 2020 (one month)/Verification by day of the week | | | |

The alighting-stop estimation method proposed in this study performs the estimation according to stages, as illustrated in Figure 1, and if the estimation is impossible in each stage, it proceeds to the next stage. Furthermore, the results estimated by Stages 1 and 2 (continuous travel, non-continuous and non-recurrent travel) are utilized as input data to construct pattern data in Stages 3 and 4. The alighting stop estimation method in each stage is as follows:

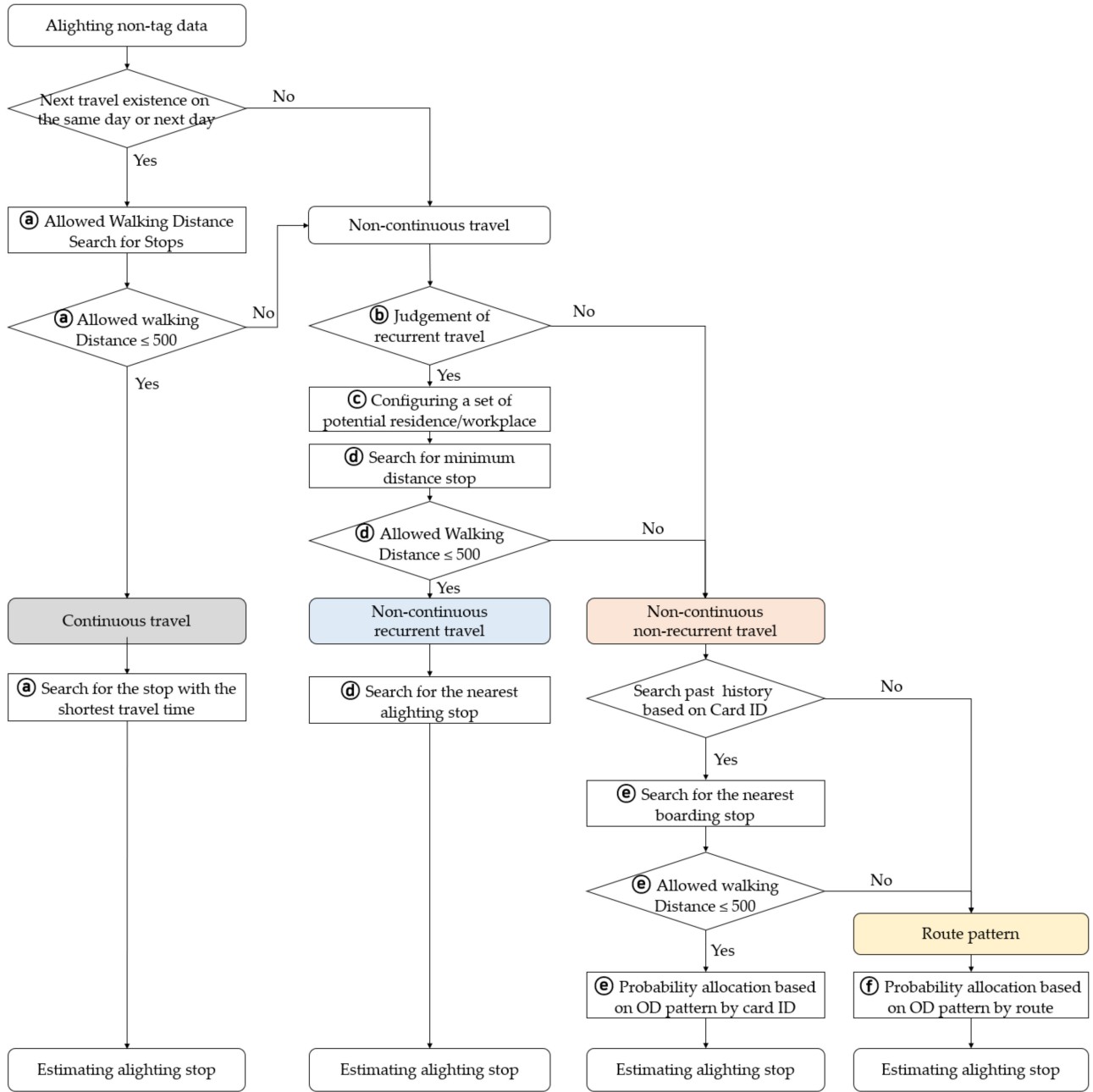

**Figure 1.** Method of gradually estimating destination of bus trip considering trip type characteristics: ⓐ Judgment of continuous travel, ⓑ Judgment of recurrent travel, ⓒ Configuring a set of potential residences/workplaces, ⓓ Judgment of non-continuous recurrent travel, ⓔ Judgment of non-continuous non-recurrent travel, ⓕ Route pattern.

Stage 1: The missing alighting stop information for "continuous travel" is estimated by applying the trip-chain-based method (see Figure 2). It is assumed that the passenger alighted the bus around the boarding stop of the immediate next trip (on the same day or the next day) of the trip that has the missing alighting information. Here, we also considered the subway station when the continuous data after the missing data of alighting the bus is pertaining to a subway. To estimate the alighting stop for continuous travel, we estimate the traveling time ($TT$) between the stops ($i$) located after the boarding stop on the route of the alighting information-missing trip and the boarding stop ($j$) of the next trip. Travel times between stops are described as follows:

$$TT_i = tt_i + wt_{i,\,j}, \forall\, i \tag{1}$$

where $TT$ is defined as the sum of the travel time ($tt_i$) from the boarding stop to the $i$-th stop and the walking time ($wt_{i,j}$) from the $i$-th stop to the $j$-th stop. Walking time is described as follows:

$$wt_{i,\,j} = \frac{\sqrt{\left(P_{j,\,x} - P_{i,\,x}\right)^2 + \left(P_{j,\,y} - P_{i,\,y}\right)^2}}{ws}, \forall i \tag{2}$$

where $wt_{i,j}$ is walking time, calculated by dividing the distance between the stops by the average walking speed ($ws$= 4 km/h). For the calculation of the distance ($d$) between the stops, we utilize the Euclidean distance, which is calculated based on the latitude coordinate ($P_x$) and longitudinal coordinate ($P_y$) data of the stops. Here, the distance between the stops $i$ and $j$ is calculated considering the bus-stop stopping order and direction for each route. Then, the estimation of the alighting stop is described as follows:

$$S_i = \min TT_i, s.t. \ d < d_l (assumed\ 500\ m) \tag{3}$$

where $S_i$, the alighting stop, is finally estimated. Specifically, the stop that minimizes the travel time is selected as the alighting stop, and the stops within the allowed walking distance ($d_l$) for the distance between the stops are targeted. To determine the spatial continuity, we classify a trip as continuous travel, only when there is a stop on the utilized route of the target trip within a certain distance from the boarding stop of the next trip. According to the result of sensitivity analysis on the allowed walking distance in a previous study, the change in the success rate and accuracy of the estimation is not significant when it is approximately 500 m [10]. In a few cases, the allowed walking distance is set to 500 m, considering that the common radius of a subway station area is 500 m, or it is set to 500–1000 m based on the distance distribution between the bus stops in the analyzed area [9,17,25,30]. In this study, we set the allowed walking distance to 500 m, considering the aforementioned studies.

① Extraction of passenger boarding stop information

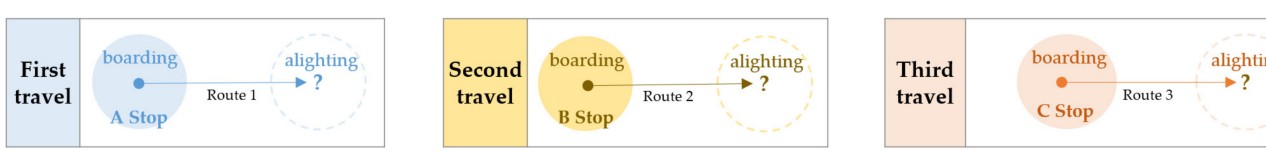

② Estimation of the alighting stop based on the next travel boarding stop

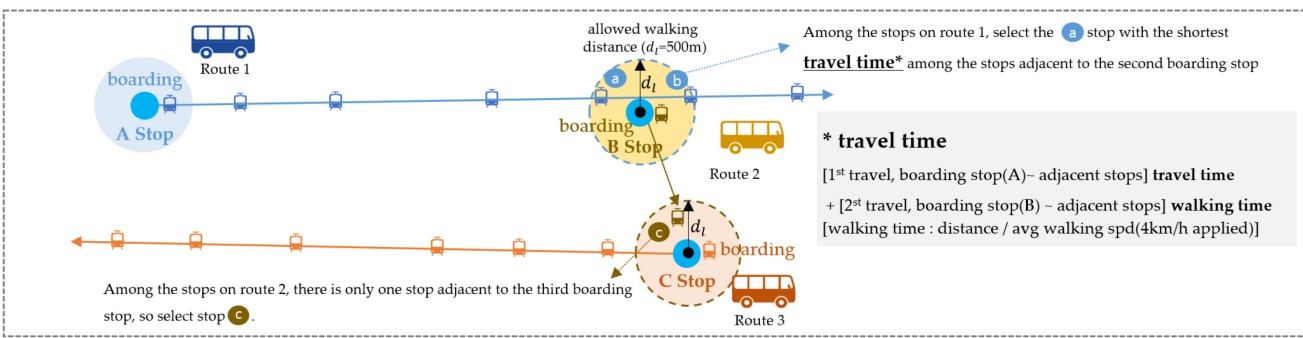

**Figure 2.** Method of estimating the alighting stop for continuous travel.

Stage 2: To estimate the alighting stop for a "recurrent travel" among the "non-continuous travel," analysis is performed first for the past travel history of each person (see Figure 3). Recurrent travel is mainly commuting trips of moving from the residence to the workplace, and back. Potential workplaces and residences are selected by solely utilizing the boarding information for each card ID to build the recurrent pattern data. To determine the recurrence of the trips, we analyzed one-month boarding data and counted the boarding frequency for each boarding stop in each time range for each card ID. To

determine the residence and workplace from the data on recurrent travel, we established the "morning commute hours" and "afternoon commute hours." The time range was set broadly to reflect various commute patterns: 04:00–12:00 for the morning commute hours and from 15:00 to 02:00 on the next day for the afternoon commute hours. Moreover, we selected the stop where the passenger boarded three times or more weekly on average in each time range as the stop at a residence or workplace candidate. The boarding stops between 04:00 and 12:00 are classified as residence candidates, and the boarding stops between 15:00 and 02:00 on the next day are classified as workplace candidates [31].

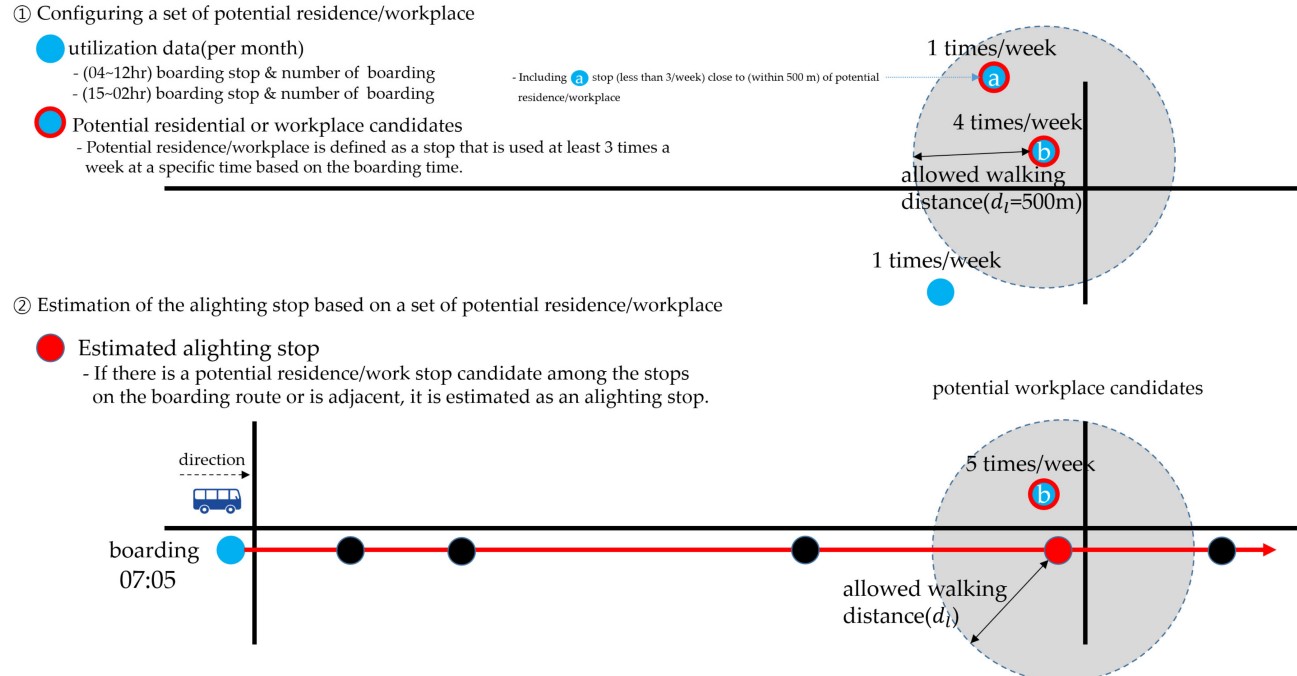

**Figure 3.** Method of estimating the alighting stop for non-continuous recurrent travel.

A stop that satisfies the criteria of a certain boarding frequency (three or more weekly) among the boarding stops in the morning commute hours is selected as a "potential residence,"; based on the same method, the boarding stop in the afternoon commute hours is selected as a "potential workplace." As for the criteria for determining recurrent travels, we selected three times or more per week by referencing a case of mobile data-based staying purpose classification (Song and Lee, 2018), which is similar in the aspect of determining travel purposes and public transportation data. Here, we also included the potential stop candidates and stops existing within 500 m, among the stops utilized less than three times per week. This also includes the cases of commuting utilizing other routes nearby.

This task is to examine the stops that have recurrent patterns based on the boarding stop information, and it is not to determine the residence and workplace accurately. In the alighting stop estimation based on the travel patterns of each person, the utilization time of the alighting stop information-missing trip is checked; the distance ($d$) is calculated between the stops located on the utilized route of the alighting information-missing trip and the stop at the "potential residence or workplace candidates"; a stop that minimizes the distance while locating with the allowed walking distance ($d_l$) is estimated to be the alighting stop. Here, regarding trips during the morning commute hours, stops are searched around the stop at the potential workplace, and regarding the afternoon commute hours, stops are searched around the stop at the potential residence.

Stage 3: Regarding "non-recurrent travel" among "non-continuous travels," the alighting stop is estimated by searching the past travel history and determining the trip that is similar to the alighting information-missing trip. As for the travel history of each card ID,

we build an alighting stop set for each boarding stop of non-recurrent trips (less than three times weekly) by utilizing the data that contain both boarding and alighting information. The non-recurrent travel pattern data are built by classifying the weekdays and weekends because the difference in the patterns is expected to be large between each day of the week. Based on the OD pattern data for each card ID, we utilize the percentage of each alighting stop for each boarding stop to estimate the alighting stop. Here, if the boarding stop is not determined in the pattern data, we search stops within a short distance (within 500 m) to determine whether the pattern data exist or not, and if there is another route, the corresponding pattern data are utilized to estimate the alighting stop based on a similar method. If there is no pattern data, we proceed to the next stage.

Stage 4: Regarding "route pattern," if it is impossible to estimate the alighting stop in Stages 1–3, we estimate it based on the OD pattern (one-year basis) for each route. This is to build 100% valid data for public transportation status analysis and utilization.

## 5. Results

### 5.1. Analysis Overview

We conducted an analysis utilizing the transportation card data of the Seoul metropolitan area, in which there is approximately no missing alighting tag information, to apply the proposed trip type classification and alighting stop estimation methods and examine the accuracy. The time range of the utilized data was one year in 2020. For the validation of the alighting stop estimation methodology developed in this study, we utilized one-month (April 2020) data of the Seoul metropolitan sphere (Seoul, Gyeonggi Province, and Incheon) where the non-tagged alighting ratio was low. As the non-tagged alighting ratio is very low in the case of subways, we solely selected the bus use data as the estimation-target. First, we built complete data (data with complete boarding-alighting records) for the accuracy validation of the alighting stop estimation method and deleted all alighting information from the data. Then, we tested the model's accuracy and identified the valid data ratio through the alighting stop estimation in each stage considering the trip type based on the data. Furthermore, because the travel patterns may be significantly different between each day of the week, we also performed the validation for each day of the week.

### 5.2. Classification Result of the Trip Types

We classified the bus trips made in April 2020 (210,920,089 trips/month) in the Seoul metropolitan area into four types (continuous, non-continuous and recurrent, non-continuous and non-recurrent, and route pattern). According to the analysis results, approximately 67.68% of the bus trips are classified as continuous travels. Table 4 shows the result of the analysis.

**Table 4.** Result of travel type classification (metropolitan area, as of April 2020).

| Division | | Number of Data | Ratio |
|---|---|---|---|
| Continuous travel | | 142,743,224 | 67.68% |
| Non-continuous | recurrent travel | 13,377,146 | 6.34% |
| | non-recurrent travel | 41,760,849 | 19.80% |
| Route pattern | | 13,038,870 | 6.18% |
| Total | | 210,920,089 | 100.00% |

Non-continuous travels are classified into recurrent and non-recurrent travels. We utilized the pattern data (weekdays/weekends) built based on the one-year boarding/alighting OD data for each card ID. As a result of classifying the trip type, 13,377,146 (6.34%) trips were counted as non-continuous recurrent travels. When no candidate stops of residence and workplace were selected, the non-continuous travels were classified as non-recurrent travels (19.80%).

Regarding route patterns, the one-year OD patterns for each route ID were constructed for all trips that could not be estimated as continuous/non-continuous trips, and the alighting stop percentages were calculated based on the boarding stop to assign the alighting stop based on the percentage. Here, we considered the moving direction on the route and assigned the alighting stop based on the probability of the corresponding stop after the route stop. Accordingly, 13,038,870 (6.18%) trips were counted as route patterns.

### 5.3. Estimated Results of Destination by Trip Type Characteristics

We calculated the estimation rate and accuracy for each trip type to validate the effectiveness of the methodology proposed in this study. The estimation rate refers to the percentage of the trips for which we can estimate the alighting stops satisfying the alighting stop estimation criteria for each stage. For the accuracy, we analyzed whether the estimated alighting stops were similar to the actual alighting stops, respectively, or comparatively analyzed the results according to the straight distance (0–1500 m) between the estimated and actual alighting stops (see Table 5). Lastly, we performed the validation for each day of the week to determine whether the travel patterns in each day of the week (weekdays/weekends) were appropriately reflected in the estimation (see Figure 4).

**Table 5.** Estimated results of destination by trip type characteristics.

| Division | | Continuous Travel | Non-Continuous Recurrent Travel | Non-Continuous Non-Recurrent Travel | Route Pattern | Total |
|---|---|---|---|---|---|---|
| Number of data | | 142,743,224 | 13,377,146 | 41,760,849 | 13,038,870 | 210,920,089 |
| Ration | | 67.68% | 6.34% | 19.80% | 6.18% | 100.00% |
| Same | matches | 85,533,555 | 5,657,869 | 11,947,068 | 1,606,002 | |
| | accuracy | 59.92% | 42.30% | 28.61% | 12.32% | 49.66% |
| | matches (accumulate) | 85,533,555 | 91,191,424 | 103,138,492 | 104,744,494 | |
| | accuracy (accumulate) | 59.92% | 58.41% | 52.12% | 49.66% | |
| 500 m | matches | 120,740,211 | 9,325,434 | 21,519,778 | 3,576,205 | |
| | accuracy | 84.59% | 69.71% | 51.53% | 27.43% | 73.56% |
| | matches (accumulate) | 120,740,211 | 130,065,645 | 151,585,423 | 155,161,628 | |
| | accuracy (accumulate) | 84.59% | 83.31% | 76.60% | 73.56% | |
| 1000 m | matches | 134,729,489 | 10,909,099 | 27,783,953 | 5,856,696 | |
| | accuracy | 94.39% | 81.55% | 66.53% | 44.92% | 85.00% |
| | matches (accumulate) | 134,729,489 | 145,638,588 | 173,422,541 | 179,279,237 | |
| | accuracy (accumulate) | 94.39% | 93.29% | 87.64% | 85.00% | |
| 1500 m | matches | 137,472,068 | 11,488,371 | 31,148,767 | 7,449,592 | |
| | accuracy | 96.31% | 85.88% | 74.59% | 57.13% | 88.92% |
| | matches (accumulate) | 137,472,068 | 148,960,439 | 180,109,206 | 187,558,798 | |
| | accuracy (accumulate) | 96.31% | 95.41% | 91.02% | 88.92% | |

Under the assumption that the alighting tag data of the public transportation cards were 100% missing, the analysis indicated that 49.66% of the estimated stops were similar to the actual stops, and in the case of less than 1 km, the destination could be identified for 85.00%. Upon analyzing the accuracy in each stage based on a range within 1 km, we determined that the accuracy was 94.39% for the continuous travels, 81.55% for the non-continuous, non-recurrent travels, 66.53% for the non-continuous, non-recurrent travels, and 44.92% for the route patterns. The assumption of the trip-chain-based alighting stop estimation method for continuous travels, i.e., the assumption that the passenger begins the next trip around the destination of the previous trip is valid. Furthermore, it is effective to apply the trip-chain-based alighting stop estimation methods to Stage 1.

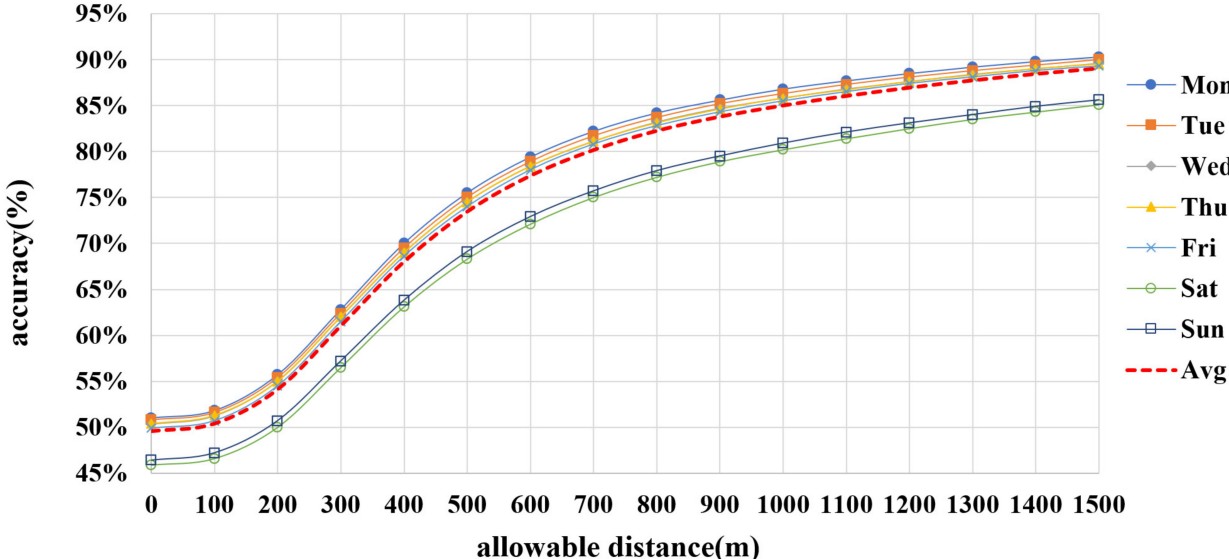

**Figure 4.** Result of destination estimation accuracy.

Non-continuous, recurrent travel can be viewed as a case where despite traveling repeatedly between certain origin and destination, the user does not travel every day, or utilizes a means other than public transportation because of a special circumstance, resulting in a disconnection between the trips. Although the percentage of the non-continuous, recurrent travels in the total bus trips is 6.34%, which is not high, this trip type can be utilized as a supplementary method to estimate the alighting stops more effectively for disconnected trips by regularity. Furthermore, because this trip type indicates better results when it is compared to the non-recurrent travels in terms of accuracy, it is appropriate to apply it to Stage 2.

For the non-continuous, non-recurrent travels, the missing alighting stops are estimated based on the past travel history (weekdays and weekends are distinguished) for each card ID. Based on the fact that the passenger's residence and major destination do not change much, even if the passenger does not travel to the destination repeatedly according to the travel history of each person, it is confirmed that this trip type can be utilized to supplement the alighting stop estimation method. However, when the error between the estimated and actual alighting stops is less than 1 km, the accuracy is 66.53%, which is low compared to that of the recurrent travels. Furthermore, among the non-continuous, non-recurrent travels, 6.18% accounted for the case where the alighting stops could not be estimated, and the estimation is impossible if there is no past travel history.

Finally, the route pattern-applied alighting stop estimation is performed when the estimation is impossible in Stages 1–3 performed earlier. According to the alighting stop estimation results, 12.32% of the estimated stops are similar to the actual alighting stops, and 44.92% accounts for the case where the distance (error) between the estimated and actual alighting stops is less than 1 km. The accuracy is relatively low because route patterns, not each person's patterns, are applied.

Upon examining the accuracy (less than 1 km) of the alighting estimation for each stage, we found that the accuracy for the continuous travels was the highest (94.39%), and that of the results utilizing route patterns was the lowest (44.92%). This study aims to ensure that more complete information is reflected when monitoring operations and making decisions by building a public transportation dataset that can be utilized in the field. Accordingly, we have designed a methodology that ensures a 100% valid tag rate by applying the route patterns in the last stage. Although the accuracy is low in the case of route patterns, there will be no challenge in utilizing them in the actual field because the composition ratio is relatively low at 6.18%.

According to the analysis on the alighting stop estimation results for each day of the week, the accuracy is low in the case of weekends (Saturday/Sunday) than in the case of weekdays. When the accuracy is examined for each stage, we find that there is no significant difference between the weekdays and the weekends in the cases of continuous travels and non-continuous, recurrent travels. However, in the cases of the non-continuous, non-recurrent travels and the route patterns, the accuracy is 2.5–5.0% lower on the weekends compared to the weekdays. The accuracy of the pattern-based estimation method is relatively low because the variability of the weekend trip patterns is relatively high.

## 6. Conclusions

This study aimed to develop a methodology for estimating non-tagged alighting stop information gradually, by considering the characteristics of trip types and utilizing transportation card data. We classified four trip types based on the continuity and recurrence of trips and developed the methodology with a trip-chain-based estimation method that considers the characteristics of individual trip type, an estimation method based on the recurrent travel patterns and non-recurrent travel patterns for each person (card ID), and finally, route patterns for building 100% valid data. The methodology proposed in this study was applied gradually to transport card data of the Seoul metropolitan area, and the estimation accuracy was examined. The results of trip type classification indicated that in approximately 67.68% of the cases, travel information existed on a similar day as the alighting information-missing trip or the next day while the trip was determined to be temporally and spatially continuous from the previous trip. Therefore, preliminarily, it was valid to apply the trip-chain-based alighting stop estimation methodology. According to the alighting stop estimation results, the accuracy (less than 1 km) of the alighting stop estimation for all samples was approximately 85.00% based on a scenario where the alighting tag data in bus trips were 100% missing, and the usable valid tag rate was close to 100%. In short, the alighting stop estimation's accuracy and valid tag rate indicated significant improvements compared to those of previous studies.

In some trips, the estimation of alighting stops was impossible based on only a single methodology, such as the trip-chain-based estimation and pattern-based estimation methodologies. In this study, we classified the trip types by analyzing the characteristics of transportation card data to build 100% complete data. Furthermore, this study differed from previous studies in that it built more complete bus trip data by applying the stage-by-stage alighting estimation model. However, on weekends, when trips tend to be less recurrent, the accuracy of estimating alighting stops was analyzed to be relatively lower compared to weekdays. This was also the case for non-continuous non-recurrent travel and route patterns for the same reason. Therefore, further studies are necessary to improve the accuracy of alighting stop estimation for weekend trips. It is also necessary to conduct additional studies to apply and review more various estimation methods such as AI-based alighting stop estimation methodology to improve the alighting estimation accuracy for non-recurrent travel. This should lead to an improvement of the current public transport OD methodology, and the discovery of public transportation policies and services.

**Author Contributions:** Conceptualization, S.L., S.C. and J.L.; methodology, S.L., S.C. and J.L.; formal analysis, S.L., S.C. and J.L.; validation, S.L., B.B. and D.N.; data curation, S.L., S.C. and J.L.; writing—original draft preparation, S.L., B.B. and D.N.; writing—review and editing, S.L., B.B. and D.N.; supervision, B.B., D.N. and S.C. All authors have read and agreed to the published version of the manuscript.

**Funding:** This paper is supported by the Korea Agency for Infrastructure Technology Advancement (KAIA) grant funded by the Ministry of Land, Infrastructure and Transport (Grant 21TLRP-B148671-04). This paper is supported by Inha University.

**Institutional Review Board Statement:** Not applicable.

**Informed Consent Statement:** Not applicpble.

**Data Availability Statement:** The data presented in this study are available on request from the corresponding author. The data are not publicly available due to privacy.

**Conflicts of Interest:** The authors declare no conflict of interest.

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
