# Peer review of "Estimating Destination of Bus Trips Considering Trip Type Characteristics"

_applsci, doi:10.3390/app112110415_

Round 1

Reviewer 1 Report

This paper presents an original methodology to estimate bus trips destinations based on transport card data. The methodology has considerable implications for governments in planning land uses and transport systems.

Nevertheless, the paper would improve by considering the following comments:

Abstract.

The abstract would benefit from a writing editing. The aim of the paper is not totally clear, so it is necessary to make more emphasis on this. Besides:

- In the sentence: “However, local governments that are applying a single-fare scheme are experiencing difficulties in using data for accurate identification of real travel patterns, policy decision support, etc.”, give at least three examples before the “etc.” or just change to “or” between the two applications.

- In line 20, when presenting the consulted references and indicating “South Korea and other countries”, it is important to be more specific. Why don’t you say at once “in different contexts” or “in Asian countries”? If it is only Asian countries, you need to justify it.

- In line 22, the sentence “This study reviewed previously conducted studies” sounds awkward.

- Please, revise the writing of this sentence “Even existing studies introduce an advanced method, we found the margin for better accuracy by combining various estimation methodologies for estimating alighting stops”. 

Please, revise the writing of the whole abstract.

Introduction

In lines 37 and 38, please, revise this sentence. It sounds awkward with many repetitions.

In line 44, I guess you mean people sometimes do not tag transport cards at the end of the trip. Terminal could mean both origin and destination. I recommend changing the word “terminal” by “at the end of the trip”.

In lines 62-63, “the proposed alighting stop estimation methodology” sounds better than the original writing.

In lines 63-64, when talking about the alighting tag rate, it would be necessary to specify if this is at entrances and/or exits.

Literature review

The literature review section would be improved by including a paragraph contextualizing this section and the parts in which it will be divided. However, due to the length of the section and the last part of implications (as a conclusion), I recommend reconsidering the section without subsections.

In lines 78-79, please revise the writing of the sentence (f. e. by changing "by linking railways and buses" by "by considering public transport routes (railways and buses)"

Line 88. Please revise the heading of section 2.2. To what type of pattern are you referring? To past travel history? Besides, probably authors should say destinations instead of destination

Line 100. Please, indicate some of the parameters of trips.

Line 101. It is recommended to use always with the same structure in the headings of subsections (methodology for estimating destinations based on...)

Line 103-104. When authors state that “there is the limitation that there are trips that are impossible to estimate using the applied methodology”, please explain why and/or give examples

Lines 113-115, Please, refer to Table 1 or some of the references included in it.

Lines 115-116. Please, clarify this statement “when a single methodology is applied, there is the limitation that the scope of the applicable trips is limited” (for instance, by adding examples)

Data.

This section should be reconsidered as Case Study, Data and Methodology.

Line 129. Please explain more in detail the case study. Authors mention “the whole country” but it is not specified in this section the case study. Besides, it is necessary to contextualize the transport system and the transportation card (mainly which transport modes work with it).

Line 134. Is it 4.1 billion/month or 4.1 billion/year?

Please, revise the headings of Table 2, regarding the editing but also the meaning: Indicate what means this ratio or what variables relate (Alighting non-tag data/total number of card validations, for instance).

Besides, it would be useful to include a map representing the regions. By including population rates in the table will also help in getting an idea of the case study.

Methodology.

First of all, one of my main concerns about the methodology is in terms of multimodality. How connections with other transport modes are considered in this scheme (my main concern is with cars)?

Methodology should also indicate more about the algorithms followed, the software used, etc.

Line 142. Revise the numbering (it should be 4.1). The same happens with the rest of the headings of this subsection

Line 143. It should be clarified if only buses are considered

Line 145. When indicating here the classification of trips, you need to, at least, enumerate here the types

Line 190. Authors need to specify the data source for obtaining the origin-destination (OD) pattern for each route

In Table 2, please use the same terminology as in the above paragraph. For instance, in “route patterns”.

Lines 200-201. It is not clear what do you mean about stages and how they appear in Figure 1. Detailed information about the stages appear latter on in the text but with no reference to figure 1.

Line 213. Please, change Euclid by Euclidean

Figure “Method of estimating the alighting stop for continuous travel” should be numbered as Figure 2. Besides, authors probably should include the radius distance in the schemes.

Line 227. Travel is an uncountable name, so it should be "non-continuous travel" or "non-continuous trips"

Results.

Line 287-288. Please revise the writing of the sentence. Probably it sounds better “we classified in four types the bus trips made in April 2020)

Lines 288-314. The information included in these lines is not part of the results but a mix of a characterization of the data and the methodology (this should appear then in the methodological section).

Line 315. According to some lines before, authors consider information from April 2020. It is not clear when saying “the one-year OD patterns” if it is a mistake or data is different for this type of trips (in this case, this should be specified in the methodology)

Results should benefit by adding a real example of a trip and the process followed to identify the alighting stop.

Reviewer 2 Report

This paper  tried to predict the alighting stop of trips in South Korea, where some trips lost that information due to the policy with a single-fare scheme adopted by local governments.

It was obvious that the authors had tried to figure above problem out and, furthermore, spent their efforts to divide those trips into four types,  "Continuous travel", "recurrent travel", "non-recurrent travel" and "route pattern". The results, the accuracy of prediction for the alighting stops, were hard to be convinced, although they also try to extend the values of  "Allowable distance" to have fault tolerance to promote the classification accuracy.

To sum up, the methods authors had tried were appreciated, but the results they achieved  were not  enough to be published. It might be a problem of data collection or cleanness that contained uncertain data. Maybe there is another point of view for these authors to verify their contribution via the differences of gain between “with” and “without” these alighting stops.

Reviewer 3 Report

The reviewed paper is dedicated to developing a methodology for estimating non-tagged alighting stop information stage-by-stage by considering the characteristics of trip types and using transportation card data. The Authors classified four trip types based on the continuity and recurrence of trips and developed the methodology with a trip-chain-based estimation method that considers the characteristics of individual trip type, an estimation method based on the recurrent travel patterns and non-recurrent travel patterns for each person, and finally, route patterns for building valid data. The paper is interesting but needs some improvements. In my opinion, the paper can be published, after taking into account the following remarks:

  • the paper text should be checked by a professional Native Speaker before printing,
  • at the end of Introduction section, the Authors wrote what was the main aim o this paper. It is good, but the Authors should also write what was contained in each paper section,
  • the literature review lacked information about the infrastructure and services surrounding public transport, such as, for example, park and ride systems or bike-sharing systems, which allow passengers to freely reach public transport stops, which significantly increases the flow and efficiency of traffic in a given area. Authors should mention about it in the literature review on the subject and thus cite the latest scientific literature in this area, i.e. in the field of park and systems e.g. "The use of a park and ride system a case study based on the City of Cracow (Poland)", doi 10.3390 / en13133473; "A Comprehensive Model to Study the Dynamic Accessibility of the Park & Ride System", doi.org/10.3390/su13074064,  etc., and in the field of bike-sharing services, e.g. "P&R parking and bike-sharing system as solutions supporting transport accessibility of the city", doi 10.21307 / TP-2020-066. One short paragraph in the Literature review section will be enough,
  • the paper text should be formatted according to the Applied Science journal requirements. Now, we can find many differences between paper text and Applied Science journal requirements,
  • the aim of this paper should be clearly presented in the introduction section and should not be repeated few times in another paper places. As far now, we can also find the aim of the paper, e.g. in "2.4. Implication" section. It should be improved,
  • the information presented in section "3. Data" is described in a very general way. One characteristic of this data is presented in Table 2. Are the Authors able to deliver more detailed data/information describing this data used for further analysis? If yes, it should be added in this section,
  • it is recommended to add below equations (1), (2), (3), word "where" and explain all variables/acronyms used in this equations. This explanation, we can also find in the paper text, but gather them below equation will improve the readability of the paper,
  • at the end of the paper, there is a lack of discussion about obtained results. We can find only the conclusion section. It is recommended to add a discussion summarized the obtained results.

Reviewer 4 Report

The authors of this paper reviewed previous studies to classify data with missing alighting stop information into trip types, and then applied a methodology to estimate alighting stops for the characteristics of each type of trip by stage. The proposed method was evaluated using transport card data from the Seoul metropolitan area and the precision was verified for each acceptable error standard for sensitivity analysis. 

The article is interesting and well structured. A language check and editorial corrections are needed (e.g. year 202 or improving the style of references: (Alsger et al., 2016, Nunes et al., 2016, 297 Gordon et al., 2013, Yang et al., 2019)). 

Round 2

Reviewer 1 Report

Dear authors,

The paper has considerably been improved and considered most of the comments suggested in previous revision.

However in lines 46-47, it is not clear why South Korea and other countries are analysed (which other countries). 

In line 56, the main porpose of the paper is not really to build reliable OD data but to suggest a methodology for estimating alighting stops 

Authors have not clarified how multimodal trips could be tackled with this methodology. If it is not possible, an explanation of this limitation should appear in the conclusions
